# Bayesian Residual Policy Optimization:
## Scalable Bayesian Reinforcement Learning with Clairvoyant Experts

## Abstract

Informed and robust decision making in the face of uncertainty is critical for robots that perform physical tasks alongside people. We formulate this as a Bayesian Reinforcement Learning problem over latent Markov Decision Processes (MDPs). While Bayes-optimality is theoretically the gold standard, existing algorithms do not scale well to continuous state and action spaces. We propose a scalable solution that builds on the following insight: in the absence of uncertainty, each latent MDP is easier to solve. We split the challenge into two simpler components. First, we obtain an ensemble of clairvoyant experts and fuse their advice to compute a baseline policy. Second, we train a Bayesian residual policy to improve upon the ensemble's recommendation and learn to reduce uncertainty. Our algorithm, Bayesian Residual Policy Optimization (BRPO), imports the scalability of policy gradient methods as well as the initialization from prior models. BRPO significantly improves the ensemble of experts and drastically outperforms existing adaptive RL methods.

## 1 Introduction

Robots operating in the real world must resolve uncertainty on a daily basis. Often times, a robot is uncertain about how the world around it evolves. For example, a self-driving car must drive safely around unpredictable actors like pedestrians and bicyclists. A robot arm must reason about occluded objects when reaching into a cluttered shelf. On other occasions, a robot is uncertain about the task it needs to perform. An assistive home robot must infer a human's intended goal by interacting with them. Both examples of uncertainty require simultaneous inference and decision making, which can be framed as Bayesian reinforcement learning (RL) over latent Markov Decision Processes (MDPs). Agents do not know which latent MDP they are interacting with, preventing them from acting optimally with respect to that MDP. Instead, *Bayes optimality* only requires that agents be optimal with respect to their current uncertainty over latent MDPs.

The Bayesian RL problem can be viewed as solving a large continuous belief MDP, which is computationally infeasible to solve directly (Ghavamzadeh et al., 2015). We build upon a simple yet recurring observation (Osband et al., 2013; Kahn et al., 2017; Choudhury et al., 2018): while solving the belief MDP may be hard, solving individual latent MDPs is much more tractable. Given exact predictions for all actors, the self-driving car can invoke a motion planner to find a collision-free path. The robot arm can employ an optimal controller to dexterously retrieve an object given exact knowledge of all objects. Once the human's intended goal is discovered, the robot can provide assistance. Hence, the overall challenge boils down to solving two (perhaps) simpler sub-challenges: solving the latent MDPs and combining these solutions to solve the belief MDP.

Let's assume we can approximately solve the latent MDPs to create an ensemble of policies as shown in Figure 1. We can think of these policies as *clairvoyant experts*, i.e., experts that think they know the latent MDP and offer advice accordingly. A reasonable strategy is to weigh these policy proposals by the belief and combine them into a single recommendation to the agent. While this recommendation is good for some regimes, it can be misleading when uncertainty is high. The onus then is on the agent to disregard the recommendation and explore the space effectively to collapse uncertainty. This leads to our key insight.

> Learning Bayesian corrections on top of clairvoyant experts is a scalable strategy for solving complex reinforcement learning problems.

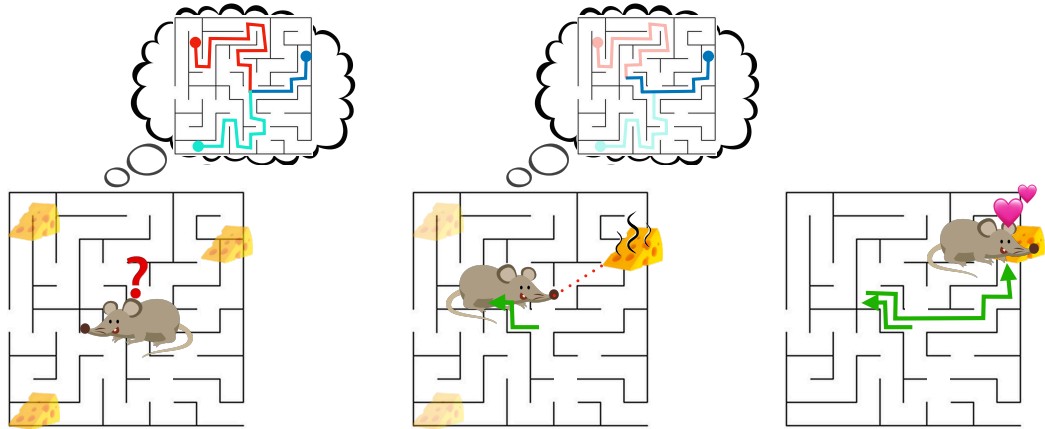

Figure 1: An overview of Bayesian Residual Policy Optimization. (a) Cheese location is unknown and expert proposals disagree about which direction to move in. (b) The Bayesian residual policy learns to smell for cheese, reducing uncertainty. (c) The experts' recommendation guides the agent to the cheese! BRPO balances exploration (Bayesian residual policy) with exploitation (expert recommendations).

While learning corrections echoes the philosophy of *boosting* (Freund & Schapire, 1999), our agent goes one step beyond: it learns to take uncertainty-reducing actions that highlight which expert to boost.

Our algorithm, Bayesian Residual Policy Optimization (BRPO), augments a belief-space batch policy optimization algorithm (Lee et al., 2019) with clairvoyant experts (Figure 1). The agent observes the experts' recommendation, belief over the latent MDPs, and state. It returns a correction over the expert proposal, including uncertainty-reducing sensing actions that experts never need to take.

Our key contribution is the following:

- We propose a scalable Bayesian RL algorithm to solve problems with complex latent rewards and dynamics.
- We experimentally demonstrate that BRPO outperforms both the ensemble of experts and existing adaptive RL algorithms.

## 2 RELATED WORK

**Belief-Space RL Methods** Bayesian reinforcement learning formalizes RL where one has a prior distribution over possible MDPs (Ghavamzadeh et al., 2015; Shani et al., 2013). However, the Bayes-optimal policy, which is the best one can do under uncertainty, is intractable to solve for and approximation is necessary (Hsu et al., 2008). One way is to approximate the value function, as done in SARSOP (Kurniawati et al., 2008) and PBVI (Pineau et al., 2003); however, they cannot deal with continous state actions. Another strategy is to resort to sampling, such as BAMCP (Guez et al., 2012), POMCP (Silver & Veness, 2010), POMCPOW (Sunberg & Kochenderfer, 2018). However, these approaches require a significant amount of online computation.

Online approaches forgo acting Bayes-optimally right from the onset, and instead aim to eventually act optimally. The question then becomes: how do we efficiently gain information about the test time MDP to act optimally? BEB (Kolter & Ng, 2009) and POMDP-lite (Chen et al., 2016) introduce an auxiliary reward term to encourage exploration and prove Probably-Approximately-Correct (PAC) optimality. This has inspired work on more general, non-Bayesian curiosity based heuristics for reward gathering (Achiam & Sastry, 2017; Burda et al., 2018; Pathak et al., 2017; Houthooft et al., 2016). Online exploration is also well studied in the bandit literature, and techniques such as posterior sampling (Osband et al., 2019) bound the learner's regret. UP-OSI (Yu et al., 2017) predicts the most likely MDP and maps that to an action. Gimelfarb et al. (2018) learns a gating over multiple expert value functions. However, online methods can over-explore and drive the agent to unsafe regimes.

Another alternative is to treat belief MDP problems as a large state space that must be compressed. Peng et al. (2018) use Long Short-Term Memory (LSTM) (Hochreiter & Schmidhuber, 1997) to encode a history of observations to generate an action. Methods like BPO (Lee et al., 2019) explicitly utilize the belief distribution and compress it to learn a policy. The key difference between BRPO and BPO is that BRPO uses an expert, enabling it to scale to handle complex latent tasks that may require multimodal policies.

**Meta-reinforcement Learning**  Meta-reinforcement learning (MRL) approaches train sample-efficient learners by exploiting structure common to a distribution of MDPs. For example, MAML (Finn et al., 2017) trains gradient-based learners while RL2 (Duan et al., 2016) trains memory-based learners. While meta-supervised learning has well established Bayesian roots (Baxter, 1998; 2000), it wasn't until recently that meta-reinforcement learning was strongly tied to Bayesian Reinforcement Learning (BRL) (Ortega et al., 2019; Rabinowitz, 2019). Nevertheless, even non-Bayesian MRL approaches address problems pertinent to BRL. MAESN (Gupta et al., 2018) learns structured noise for exploration. E-MAML (Stadie et al., 2018) adds an explicit exploration bonus to the MAML objective. GMPS (Mendonca et al., 2019) exploit availability of MDP experts to partially reduce BRL to IL. Our work is more closely related to Bayesian MRL approaches. MAML-HB (Grant et al., 2018) casts MAML as hierarchical Bayes and improves posterior estimates. BMAML (Yoon et al., 2018) uses non-parametric variational inference to improve posterior estimates. PLATIPUS (Finn et al., 2018) learns a parameter distribution instead of a fixed parameter. PEARL (Rakelly et al., 2019) learns a data-driven Bayes filter across tasks. In contrast to these approaches, we use experts at test time, learning only to optimally correct them.

**Residual Learning**  Residual learning has its foundations in boosting (Freund & Schapire, 1999) where a combination of weak learners, each learning on the failures of previous, make a strong learner. It also allows for injecting priors in RL, by boosting off of hand-designed policies or models. Prior work has leveraged known but approximate models by learning the residual between the approximate dynamics and the discovered dynamics (Ostafew et al., 2014; 2015; Berkenkamp & Schoellig, 2015). There has also been work on learning residual policies over hand-defined ones for solving long horizon (Silver et al., 2018) and complex control tasks (Johannink et al., 2019). Similarly, our approach starts with a useful initialization (via experts) and learns to improve via Bayesian reinforcement learning.

## 3  PRELIMINARIES: BAYESIAN REINFORCEMENT LEARNING

In Bayesian reinforcement learning, the agent does not know the reward and transition functions but knows that they are determined by a latent variable $\phi \in \Phi$. Formally, the problem is defined by a tuple $\langle S, \Phi, A, T, R, P_0, \gamma \rangle$, where $S$ is the observable state space of the underlying MDPs, $\Phi$ is the latent space, and $A$ is the action space. $T$ and $R$ are the transition and reward functions parameterized by $\phi$. The initial distribution over $(s, \phi)$ is given by $P_0 : S \times \Phi \to \mathbb{R}^+$, and $\gamma$ is the discount.

Bayesian RL considers the long-term expected reward with respect to the uncertainty over $\phi$ rather than the true (unknown) value of $\phi$. Uncertainty is represented as a *belief distribution* $b \in B$ over latent variables $\phi$. The Bayes-optimal action value function is given by the Bellman equation:

$$Q(s, b, a') = R(s, b, a') + \gamma \sum_{s', b'} P(s'|s, b, a') P(b'|s, b, a') \max_{a''} Q(s', b', a'') \tag{1}$$

The Bayesian reward function is the expected reward $R(s, b, a') = \sum_{\phi \in \Phi} b(\phi) R(s, \phi, a')$. The Bayesian transition function is $P(s'|s, b, a') = \sum_{\phi \in \Phi} b(\phi) P(s'|s, \phi, a')$. The posterior update $P(b'|s, b, a')$ is computed recursively, starting from initial belief $b_0$.

$$b'(\phi'|s, b, a', s') = \eta \sum_{\phi \in \Phi} b(\phi) T(s, \phi, a', s', \phi') \tag{2}$$

where $\eta$ is the normalizing constant, and the transition function is defined as $T(s, \phi, a', s', \phi') = P(s', \phi'|s, \phi, a') = P(s'|s, \phi, a') P(\phi'|s, \phi, a', s')$. At timestep $t$, the belief $b_t(\phi_t)$ is the posterior distribution over $\Phi$ given the history of states and actions, $(s_0, a_1, s_1, ..., s_t)$. When $\phi$ corresponds to physical parameters for an autonomous system, we often assume that the latent states are fixed.

---

**Algorithm 1** Bayesian Residual Policy Optimization

---

**Require:** Bayes filter $\psi$, belief $b_0$, prior $P_0$, residual policy $\pi_{\theta_0}$, expert $\pi_e$, horizon $H$, $n_{\text{itr}}$, $n_{\text{sample}}$

1: **for** $i = 1, 2, \cdots, n_{\text{itr}}$ **do**
2:      **for** $n = 1, 2, \cdots, n_{\text{sample}}$ **do**
3:          Sample latent MDP $M$: $(s_0, \phi_0) \sim P_0$
4:          $\tau_n \leftarrow \text{Simulate}(\pi_{\theta_{i-1}}, \pi_e, b_0, \psi, M, H)$
5:      Update policy: $\theta_i \leftarrow \text{BatchPolicyOptimization}(\theta_{i-1}, \{\tau_1, \cdots, \tau_{n_{\text{sample}}}\})$
6: **return** $\pi_{\theta_{best}}$

7: **procedure** SIMULATE($\pi_\theta, \pi_e, b_0, \psi, M, H$)
8:      **for** $t = 1, \cdots, H$ **do**
9:          $a_{e_t} \sim \pi_e(\cdot | s_{t-1}, b_{t-1})$ // Expert recommendation
10:          $a_t \leftarrow a_{r_t} + a_{e_t}, \quad a_{r_t} \sim \pi_\theta(s_{t-1}, b_{t-1}, a_{e_t})$ // Residual action
11:          Execute $a_t$ on $M$, observe $r_t, s_t$
12:          $b_t \leftarrow \psi(s_{t-1}, b_{t-1}, a_t, s_t)$
13:      $\tau \leftarrow (s_0, b_0, a_{r_1}, r_1, s_1, b_1, \cdots, a_{r_H}, r_H, s_H, b_H)$ // Only residuals are recorded
14:      **return** $\tau$

---

Our algorithm utilizes a black-box Bayes filter to produce a posterior distribution over the latent states. However, a Bayes filter can also be interpreted as a function that compresses the history of states and actions. Recent work suggests that Long Short-Term Memory (LSTM) cells (Hochreiter & Schmidhuber, 1997) can be meta-trained to compress history and predict subsequent states (Ortega et al., 2019). Such learned representations can be substituted for the belief distribution that we have chosen here.

## 3.1 BAYESIAN REINFORCEMENT LEARNING AND POSTERIOR SAMPLING

Posterior Sampling Reinforcement Learning (PSRL) (Osband et al., 2013) is an online RL algorithm that maintains a posterior over latent MDP parameters $\phi$. However, the problem setting it considers and how it uses this posterior are quite different than what we consider in this paper.

In this work, we are focused on zero-shot scenarios where the agent can only interact with the test MDP for a single episode; latent parameters are resampled for each episode. The PSRL regret analysis assumes MDPs with finite horizons and repeated episodes with the same test MDP, i.e. the latent parameters are fixed for all episodes.

Before each episode, PSRL samples an MDP from its posterior over MDPs, computes the optimal policy for the sampled MDP, and executes it on the fixed test MDP. Its posterior is updated after each episode, concentrating the distribution around the true latent parameters. During this exploration period, it can perform arbitrarily poorly (see Section 6.1 of the appendix for a concrete example). Furthermore, sampling a latent MDP from the posterior determinizes the parameters; as a result, there is no uncertainty in the sampled MDP, and the resulting optimal policies that are executed will never take sensing actions. In this work, we have focused on Bayesian RL problems where sensing is critical to performance. BRPO, like other Bayesian RL algorithms, focuses on learning the Bayes-optimal policy during training, which can be used at test time to immediately explore and exploit in a new environment.

## 4 BAYESIAN RESIDUAL POLICY OPTIMIZATION (BRPO)

In Bayesian Residual Policy Optimization, we first construct an ensemble of clairvoyant experts where each approximately solves a latent MDP. Expert proposals are gated with the belief over MDPs to compute a recommendation. We then train a Bayesian residual policy to correct the recommendation, resulting in an elegant exploration-exploitation tradeoff. The agent learns to produce smaller corrections when the recommendation is effective, i.e. when uncertainty is small or when all clairvoyant experts agree with a good recommendation. Otherwise, the agent overrides the recommendation and learns to explore effectively.

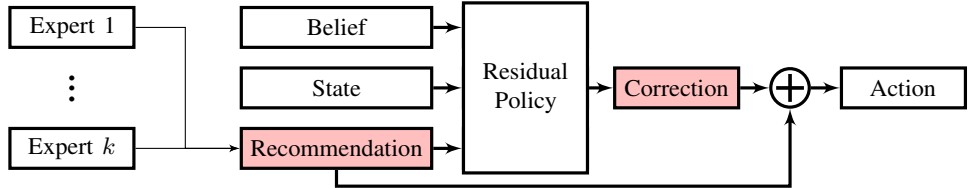

Figure 2: Bayesian residual policy network architecture.

## 4.1 ENSEMBLE OF CLAIRVOYANT EXPERTS

For simplicity of exposition, assume the Bayesian RL problem consists of $k$ underlying latent MDPs, $\phi_1, ..., \phi_k$. Clairvoyant experts $\pi_i$ can be computed for each $\phi_i$ via single-MDP RL methods (or optimal control, if transition and reward functions are known). If $b(\phi_i)$ is the posterior belief over each MDP $\phi_i$, we want to combine experts to construct a belief-aware recommendation that maps the state and belief to a distribution over actions $\pi_e : S \times B \to P(A)$.

One choice for $\pi_e$ is to select the maximum a posteriori (MAP) action.

$$a_{\text{MAP}} = \arg\max_a \sum_{i=1}^{k} b(\phi_i)\pi_i(a|s) \tag{3}$$

However, computing the MAP estimate may require optimizing a non-convex function, e.g., when the distribution is multimodal. We can instead maximize the lower bound using Jensen's inequality.

$$\log \sum_{i=1}^{k} b(\phi_i)\pi_i(a|s) \geq \sum_{i=1}^{k} b(\phi_i)\log \pi_i(a|s) \tag{4}$$

This is much easier to solve, especially if $\log \pi_i(a|s)$ is convex. If each $\pi_i(a|s)$ is a Gaussian with mean $\mu_i$ and covariance $\Sigma_i$, e.g. from TRPO (Schulman et al., 2015), the resultant action is

$$a^* = \arg\max_a \sum_{i=1}^{k} b(\phi_i)\log \pi_i(a|s) = \left[\sum_{i=1}^{k} b(\phi_i)\Sigma_i^{-1}\right]^{-1} \sum_{i=1}^{k} b(\phi_i)\Sigma_i^{-1}\mu_i \tag{5}$$

When the belief has collapsed to one $\phi_i$, the resulting ensemble recommendation follows the corresponding $\pi_i$ exactly. Thus, as entropy reduces, the ensemble is more reliable.

There are other alternatives to consider. One choice for $\pi_e$ is to directly use the mixture model $\sum_{i=1}^{k} b(\phi_i)\pi_i(a|s)$. This would be equivalent to posterior sampling (Osband et al., 2013).

While this belief-aware ensemble is easy to attain, it is not Bayes-optimal. In particular, since the clairvoyant experts do not take explicit uncertainty-reducing actions, the ensemble will not recommend to do so. Consider the `Maze4` example: each clairvoyant expert knows its corresponding latent MDP's hidden goal position, and thus navigates optimally without sensing. A Bayes-optimal agent, on the other hand, would take sensing actions to identify the latent goal.

## 4.2 RESIDUAL POLICY LEARNING

In each training iteration, BRPO collects trajectories by simulating the current policy on several MDPs sampled from the prior distribution (Algorithm 1). At every timestep of the simulation, the ensemble is queried for an action recommendation, which is summed with the correction from the residual policy network (Figure 2) and executed. The Bayes filter updates the posterior after observing the resulting state. The collected trajectories (with only residual actions) are the input to a policy optimization algorithm (Schulman et al., 2015; 2017) which updates the residual policy network.

The residual policy does not solve the original belief MDP. Since its corrective actions are summed with the ensemble's recommendations, it in fact operates in a residual belief MDP (with respect to the recommendations). Actions are simply shifted by the recommendations. That is, for every residual

action $a_{r'}$ and expert recommendation $a_{e'} \sim \pi_e(\cdot|s, b)$, we can define a new transition dynamics $\tilde{T}$ from the original $T$:

$$P_{\tilde{T}}(s', b', r'|s, b, a_{r'}) = \int_{a_{e'} \in A} P_T(s', b', r'|s', b', a_{r'} + a_{e'}) \pi_e(a_{e'}|s, b) \quad (6)$$

This new transition function $\tilde{T}$ defines the residual belief MDP. Since the experts are fixed, the residual belief-MDP is also fixed during training and testing time. Thus, this strategy inherits all mathematical guarantees from the underlying policy optimization algorithm, such as monotonic improvement from the ensemble's baseline policy. This leads to the following theorem.

**Theorem 1.** BRPO *inherits the mathematical properties of the underlying batch policy optimization algorithm.*

*Proof.* The proof directly follows from Equation 6. $\qquad\square$

## 5 Experimental Results

We choose problems that highlight common challenges in robotics:

- Explicit or implicit sensing actions are required to infer the latent MDP.
- Sensing is costly, and different sensing actions may have different costs.
- Solutions for each latent MDP are significantly different.

In all domains that we consider, BRPO improves on the ensemble's recommendation, learning to sense in a cost-effective manner. As seen in `Door4`, BRPO also can develop more effective strategies for both sensing and control.

**Latent Goal Mazes** In the **Maze4** and **Maze10** environments, the agent must identify which latent goal is active. This is an example of where the dynamics are the same across all latent MDPs, but the task must be inferred.

At the beginning of each episode, the latent goal is set to one of four goals (`Maze4`) or ten goals (`Maze10`). This problem requires explicit sensing to distinguish the goal. Sensing can happen simultaneously as the agent moves, but costs $-1$; the agent receives a noisy measurement of the distance to the goal, with noise proportional to the true distance. This motivates the agent to minimize sensing and sense when closer to goals to obtain better measurements.

After each action, the agent observes its current position, velocity, and distance to all latent goals. If sensing is invoked, it also observes the noisy distance to the goal. In addition, the agent observes the categorical belief distribution over the latent goals and the ensemble's recommendation. Each expert proposes an action (computed via motion planning) that navigates to the corresponding goal. The experts are unaware of the penalty for reaching incorrect goals, which will demonstrate that BRPO can improve on such suboptimal experts.

In `Maze4`, reaching the active goal provides a terminal reward of $500$, while reaching an incorrect goal gives a penalty of $-500$. The task ends when the agent receives either the terminal reward or penalty, or after $500$ timesteps. In `Maze10`, the agent receives a penalty of $-50$ and continues to explore after reaching an incorrect goal.

Figure 3a demonstrates rollouts by the trained BRPO agents on `Maze4` and `Maze10`. In `Maze10`, goals that are near each other have drastically different paths to them, making task inference even more important. For both `Maze4` and `Maze10`, we see that the agent reroutes itself (multiple times in `Maze10`) while it invokes sensing to get a better belief.

**Doors** In this more classical POMDP problem, there are 4 possible doors to the goal in the next room. At the beginning of each episode, each door is opened or closed with $0.5$ probability. To check the doors, the agent can either sense ($-1$) or crash into them ($-10$). As with the mazes, the sense action can be taken simultaneously as the agent moves. Sensing returns a noisy binary vector for all four doors, with exponentially decreasing accuracy proportional to the distance to each door.

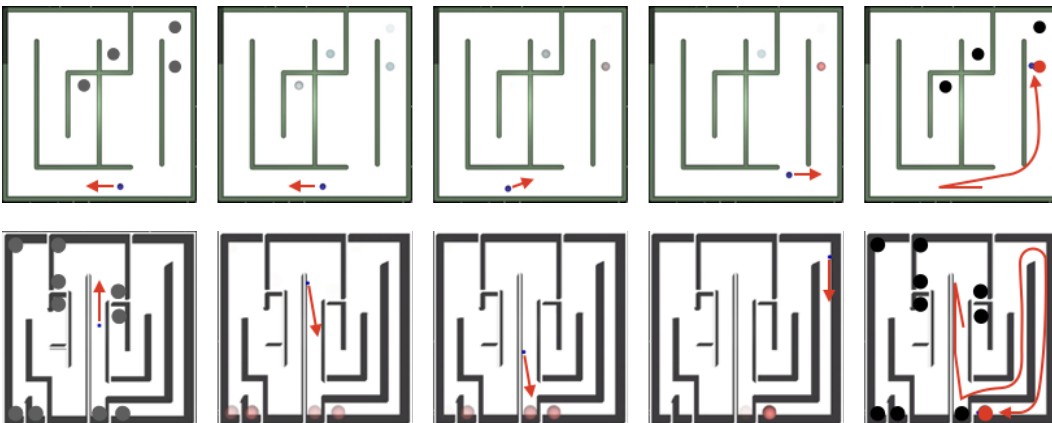

(a) Latent goal mazes with four (**Maze4**) and ten (**Maze10**) possible goals. The agent senses as it navigates, changing its direction as goals are deemed less likely (more transparent). We have marked the true goal with red in the last frame for clarity.

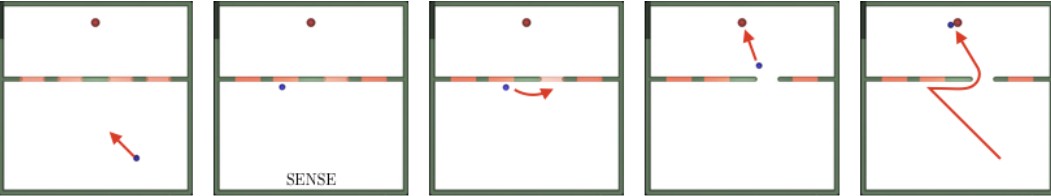

(b) **Door4**. The agent senses only when it is near the wall with doors, where sensing is most accurate. The transparency of the red bars indicates the posterior probability that the door is blocked. With sensing, the third door becomes more likely to be open while the others become more likely to be closed.

Figure 3: BRPO policy keyframes. Best viewed in color.

Crashing returns an accurate indicator of the door it crashed into. At every step, the agent observes its position, velocity, distance to goal, and whether it crashed or passed through a door. In addition, the agent observes the categorical distribution over the $2^4 = 16$ possible door configurations (from the Bayes filter) and the ensemble's recommendation. The agent receives a terminal reward of 100 if it reaches the goal within 300 timesteps.

We observe that BRPO's learned policy is quite different from any of the experts. Each expert navigates directly through the closest open door. BRPO gets very close to the wall (to minimize sensor noise) and senses while sliding along the wall, before identifying an open door and navigating through it.

## 5.1 BRPO OUTPERFORMS ADAPTIVE RL METHODS

We compare BRPO to adaptive RL algorithms that consider the belief over latent states: BPO (Lee et al., 2019) and UP-MLE, a modification to Yu et al. (2017) that uses the most likely estimate from the Bayes filter[1]. We also compare with the ensemble of experts baseline, with one key difference. The ensemble will not take any sensing actions (as discussed in Section 4), so we strengthen it by sensing with probability 0.5 at each timestep. This is equivalent to the initial BRPO policy, which adds random noise to the ensemble recommendation. More sophisticated sampling strategies can be considered but require more task-specific knowledge to design; see the appendix for more discussion (Section 6.2).

Figure 4a compares the training performance of all algorithms across the three environments. (Note that OPTIMAL is unachievable, since it requires full knowledge of the latent MDP.) In RANDOM, the agent randomly chooses one of the clairvoyant experts to follow for the entire episode.

---

[1]This was originally introduced in Lee et al. (2019).

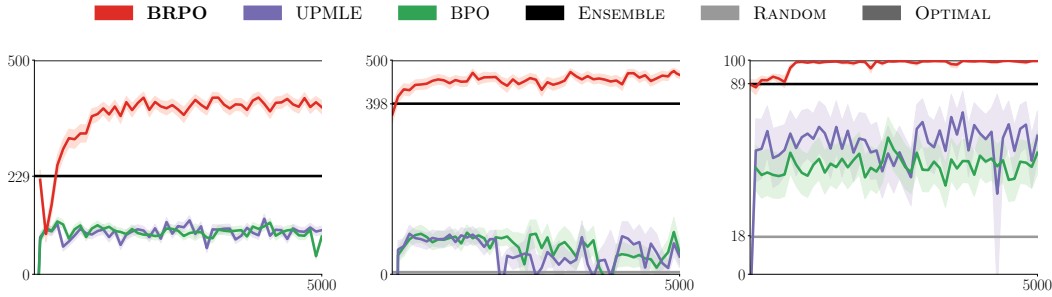

(a) Training curves. BRPO dramatically outperforms agents that do not leverage expert knowledge, and significantly improves the ensemble of experts.

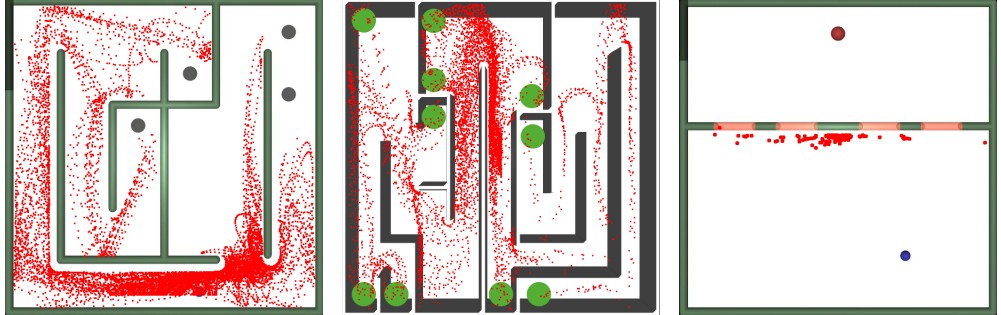

(b) Sensing locations. In `Maze4` and `Maze10`, sensing is dense around the starting regions (the bottom row in `Maze4` and center in `Maze10`) and in areas where multiple latent goals are nearby. The agent sometimes reroutes before reaching an incorrect goal. In `Door4`, BRPO only senses when close to the doors, where the sensor is most accurate.

Figure 4: BRPO performance on `Maze4`, `Maze10`, and `Door4` (left to right).

BRPO agents dramatically outperforms BPO and UP-MLE agents. In fact, we have trained BPO and UP-MLE with an additional boost to encourage information-gathering (Section 5.3); without such bonuses, they did not learn to take any meaningful behavior. Even with the bonus, these agents learns to solve the task only partially. In `Maze4` and `Maze10`, they only reach some of the goals. In `Door4`, they only learn to navigate through one of the first two doors and will occasionally crash.

Examining where sensing has happened (Figure 4b), we see that the BRPO agent learns to sense when goals must be distinguished, and uses the belief to reroute itself in `Maze4` and `Maze10`. Qualitatively, we find that UP-MLE relies exclusively on crashing into doors to reduce uncertainty, which is extremely costly. The BRPO agent avoids crashing in almost all scenarios.

## 5.2 RESIDUAL POLICY INPUTS

The BRPO policy takes the belief distribution, state, and ensemble recommendation as inputs (Figure 2). However, since the ensemble recommendation implicitly includes the belief, the belief may not be a necessary input to the policy if the recommendation is already provided.

The results show that providing both belief and recommendation as inputs to the policy are important (Figure 5a). Although BRPO with only the recommendation performs comparably to BRPO with both inputs on `Maze4` and `Maze10`, the one with both inputs produce faster learning on `Door4`.

## 5.3 INFORMATION-GATHERING REWARD BONUSES

Because BRPO maximizes the Bayesian Bellman equation (Equation 1), exploration is incorporated into its long-term objective. As a result, auxiliary rewards to encourage exploration are unneeded. However, existing work that does not explicitly consider the belief has suggested various auxiliary reward terms that encourage exploration, such as intrinsic rewards (Pathak et al., 2017) or surprisal

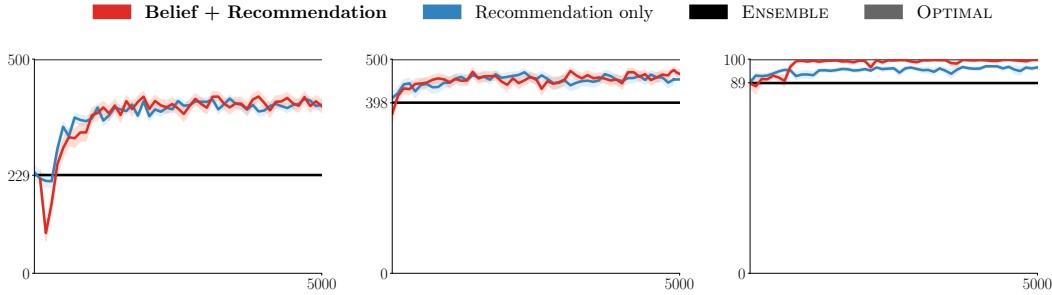

(a) Including both belief and recommendation as policy inputs results in faster learning in `Door4`.

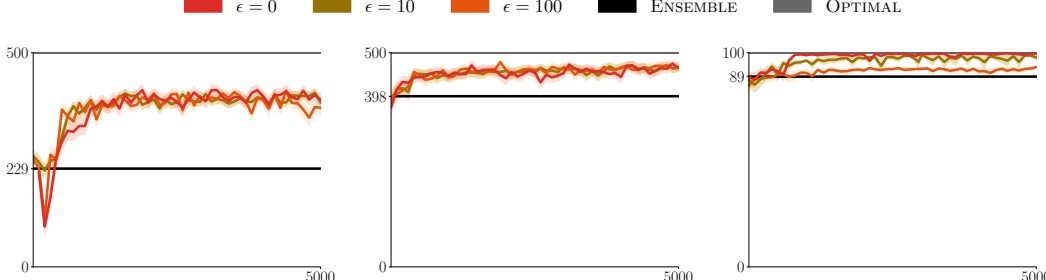

(b) Information-gathering reward bonuses (Equation 7) are unnecessary with BRPO. Large bonuses can cause the agent to ignore penalties from the environment, resulting in suboptimal performance (`Door4`).

Figure 5: Residual policy input and reward bonus experiments on `Maze4`, `Maze10`, and `Door4` (left to right).

rewards (Achiam & Sastry, 2017). To investigate whether such rewards benefit the BRPO agent, we augment the reward function with the following auxiliary bonus:

$$\tilde{r}(s_t, b_t, a_t) = r(s_t, b_t, a_t) + \epsilon \cdot \mathbb{E}_{b_{t+1}} \left[ \|b_t - b_{t+1}\|_1 \right] \tag{7}$$

where $\|b_t - b_{t+1}\|_1 = \sum_{i=1}^{k} |b_t(\phi_i) - b_{t+1}(\phi_i)|$ rewards change in belief.[2]

Figure 5b summarizes the performance of BRPO when training with $\epsilon = 0, 10, 100$. Too much emphasis on information-gathering causes the agent to over-explore and therefore underperform. In `Door4` with $\epsilon = 100$, we qualitatively observe that the agent crashes into the doors more often. This is because crashing significantly changes the belief for that door; the huge reward bonus outweighs the penalty of crashing from the environment.

We find that BPO and UP-MLE are unable to learn without an exploration bonus. We used $\epsilon = 1$ for `Maze4` and `Door4`, and $\epsilon = 100$ for `Maze10`. With the bonus, both BPO and UP-MLE learn to sense initially but struggle to solve the challenging latent MDPs.

## 6 DISCUSSION AND FUTURE WORK

In the real world, robots must deal with uncertainty, either due to complex latent dynamics or task specifics. Because uncertainty is an inherent part of these tasks, we can at best aim for optimality under uncertainty, i.e., Bayes optimality. Existing BRL algorithms or POMDP solvers do not scale well to problems with complex latent MDPs or a large (continuous) set of MDPs.

We decompose BRL problems into two parts: solving each latent MDP and being Bayesian over the solutions. Our algorithm, Bayesian Residual Policy Optimization, operates on the residual belief-MDP space given an ensemble of experts. BRPO focuses on learning to explore, relying on the experts for exploitation. BRPO is capable of solving complex problems, outperforming existing BRL algorithms and improving on the original ensemble of experts.

---

[2]An analogous term has been introduced in Chen et al. (2016).

Although out of scope for this work, a few key challenges remain. First is an efficient construction of an ensemble of experts, which becomes particularly important for continuous latent spaces with infinitely many MDPs. Infinitely many MDPs do not necessarily require infinite experts, as many may converge to similar policies. An important future direction is subdividing the latent space and computing a qualitatively diverse set of policies (Liu et al., 2016). Another challenge is developing an efficient Bayes filter, which is an active research area. In certain occasions, the dynamics of the latent MDPs may not be accessible, which would require a learned Bayes filter. Combined with a tractable, efficient Bayes filter and an efficiently computed set of experts, we believe that BRPO will provide an even more scalable solution for BRL problems.

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

APPENDIX

6.1    THE GAP BETWEEN BAYES OPTIMALITY AND POSTERIOR SAMPLING

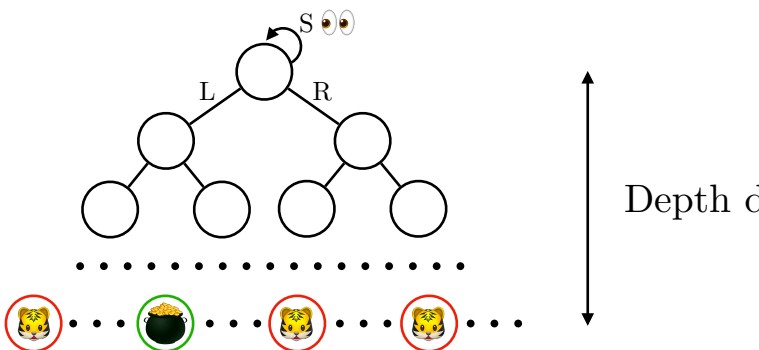

Figure 6: A tree-like MDP that highlights the distinction between BRL and PSRL.

As discussed in Section 3.1, Bayesian reinforcement learning and posterior sampling address quite different problems. We present a toy problem to highlight the distinction between them.

Consider a deterministic tree-like MDP (Figure 6). Reward is received only at the terminal leaf states: one leaf contains a pot of gold ($R = 100$) and all others contain a dangerous tiger ($R = -10$). All non-leaf states have two actions, go left (L) and go right (R). The start state additionally has a sense action (S), which is costly ($R = -0.1$) but reveals the exact location of the pot of gold. Both algorithms are initialized with a uniform prior over the $N = 2^d$ possible MDPs (one for each possible location of the pot of gold).

To contrast the performance of the Bayes-optimal policy and posterior sampling, we consider the multi-episode setting where the agent repeatedly interacts with the same MDP. The MDP is sampled once from the uniform prior, and agents interact with it for $T$ episodes. This is the setting typically considered by posterior sampling (PSRL) (Osband et al., 2013).

Before each episode, PSRL samples an MDP from its posterior over MDPs, computes the optimal policy, and executes it. After each episode, it updates the posterior and repeats. Sampling from the posterior determinizes the underlying latent parameter. As a result, PSRL will never produce sensing actions to reduce uncertainty about that parameter because the sampled MDP has no uncertainty. More concretely, the optimal policy for each tree MDP is to navigate directly to the gold *without sensing*; PSRL will never take the sense action. Thus, PSRL makes an average of $\frac{N-1}{2}$ mistakes before sampling the correct pot of gold location and the cumulative reward over $T$ episodes is

$$-10 \underbrace{\left(\tfrac{N-1}{2}\right)}_{\text{mistakes}} + 100 \underbrace{\left(T - \tfrac{N-1}{2}\right)}_{\text{pot of gold}} \tag{8}$$

In the first episode, the Bayes-optimal first action is to sense. All subsequent actions in this first episode navigate toward the pot of gold, for an episode reward of $-0.1 + 100$. In the subsequent $T - 1$ episodes, the Bayes-optimal policy navigates directly toward the goal without needing to sense, for a cumulative reward of $100T - 0.1$. The performance gap between the Bayes-optimal policy and posterior sampling grows exponentially with depth of the tree $d$.

Practically, a naïve policy gradient algorithm (like BPO) would struggle to learn the Bayes-optimal policy: it would need to learn to both sense and navigate the tree to the sensed goal. BRPO can take advantage of the set of experts, which each navigate to their designated leaf. During training, the BRPO agent only needs to learn to balance sensing with navigation.

As mentioned in Section 3.1, PSRL is an online learning algorithm and is designed to address domains where the posterior naturally updates as a result of multiple episodes of interactions with the latent MDP. PSRL is more focused on improving the performance over episodes, which is different from the average performance or zero-shot performance that we consider in this work.

## 6.2 EXPERIMENTS: BETTER SENSING ENSEMBLE

The ensemble we considered in Section 5 randomly senses with probability 0.5. A more effective sensing ensemble baseline policy could be designed manually, and used as the initial policy for the BRPO agent to improve on. Designing such a policy can be challenging: it requires either task-specific knowledge, or solving an approximate Bayesian RL problem. We bypass these requirements by using BRPO.

On the `Maze10` environment, we have found via offline tuning that a more effective ensemble baseline agent senses only for the first 150 of 750 timesteps. The average return is $416.3 \pm 9.4$, which outperforms the original ensemble baseline average return of $409.5 \pm 10.8$. However, this is still lower than the BRPO agent that starts with that original ensemble, which accumulated an average return of $465.7 \pm 4.7$. This trained BRPO agent also achieves a task completion rate of $100\%$, which is better than the $96.3\%$ completed by the improved ensemble baseline. The performance gap comes from the suboptimality of the ensemble recommendation, as experts are unaware of the penalty for reaching incorrect goals.

## 6.3 EXPERIMENTS: POSTERIOR SAMPLING

We have also evaluated the performance of PSRL on the `Maze10` environment. This has required a slight modification to PSRL to handle the zero-shot scenario: it now samples from the posterior at each timestep, and executes the corresponding optimal expert (which is aware of the penalties from reaching the wrong goal). However, PSRL never senses. As a result, this vanilla agent frequently incurs the penalty for reaching incorrect goals, achieving an average return of $-124.4 \pm 11.3$. Augmenting the PSRL agent by sensing with probability 0.5 (as with the ensemble method) results in an average return of $464.1 \pm 5.5$, and a task completion rate of $94\%$. Failures occur when the posterior does not collapse to a single target for the posterior-sampled experts to navigate toward.

