# OpenReview forum: "Bayesian Residual Policy Optimization: Scalable Bayesian Reinforcement Learning with Clairvoyant Experts"
_ICLR.cc/2020/Conference — Reject_

### Official Review · AnonReviewer1 · 2019-10-22
**Official Blind Review #1**

**Rating:** 3

**Review:**

This paper considers Bayesian Reinforcement Learning problem over latent Markov Decision Processes (MDPs). The authors consider making decisions with experts, where each expert performs well under some latent MDPs. An ensemble of experts is constructed, and then a Bayesian residual policy is learned to balance exploration-exploitation tradeoff. Experiments on Maze and Door show the advantages of residual policy learning over some baselines.

1. The Bayesian Reinforcement Learning problem this work considered is important. However, using experts immediately make the problem much easier. The original Bayesian Reinforcement Learning problem is then reduced to making decision with experts. Under this setting, there are many existing work with respect to exploration-exploitation tradeoff (OFU, Thompson Sampling) with theoretical guarantees. I did not see why using this residual policy learning (although as mentioned residual/boosting is useful under other settings) is reasonable here. There is not theoretical support showing that residual learning enjoys guaranteed performance. The motivation of introducing this heuristic is not clear.

2. The comparisons with UPMLE and BPO seems not convincing. Both BPO and UPMLE do not use experts, and ensemble of experts outperforms them as shown in the experiments. And the ensemble baseline here is kind of weak (why sensing with probability 0.5 at each timestep?) Always 0.5 does not make sense (exploration should decrease as uncertainty reduced). Other exploration methods should be compared, to empirically show the advantages/necessities of residual policy learning.

Overall, I consider the proposed BRPO a simple extension of BPO, with a heuristic of learning ensemble policy to make decisions. BRPO is lack of theoretical support, and it is not clear why residual policy learning here is necessary and what exactly the advantage is over other exploration methods. Comparisons with simple baseline like exploration with constant probability is not enough to justify the proposed method.

=====Update=====
Thanks for the rebuttal. The comparison with PSRL improves the paper. However, I still think this paper needs more improvement as follows.
Theorem 1 looks hasty to me. Batch policy optimization Alg is going to solve n_{sample} MDPs, which are generated from P_0. But Eq. (6) or Theorem 1 does not contain information about P_0, implying that P_0 has no impact, which is questionable (an uniform P_0 that can generate different MDPs and a deterministic P_0 can only generate one MDP should be very different). I suggest the authors do more detailed analysis.
On the other hand, I expected whether this special "residual action" heuristic has any guarantees in RL? Can decomposing action into a_r + a_e provide us a better exploration method (than others like PSRL, OFU...)? Since this is the main idea of this paper as an extension of BPO, I think this point is important. The experiments shows that it can work in some cases, but I do not see an explanation (the "residual learning" paragraph is high level and I do not get an insight from that.).

**Experience Assessment:**

I have published one or two papers in this area.

**Review Assessment: Checking Correctness Of Derivations And Theory:**

I carefully checked the derivations and theory.

**Review Assessment: Checking Correctness Of Experiments:**

I carefully checked the experiments.

**Review Assessment: Thoroughness In Paper Reading:**

I read the paper thoroughly.

---

> ### Author Response · Authors · 2019-11-15
> **Response**
>
> Thank you for your thoughtful comments and feedback. We have updated our paper to address your questions and comments. Here’s our summary.
>
> 1. What is the advantage of BRPO over Thompson Sampling (PSRL)?
>
> We have added Section 3.1 and Appendices 6.1 and 6.3 to discuss the distinction. Those sections are repeated below for convenience.
>
> Posterior Sampling Reinforcement Learning (PSRL) [1] is an online RL algorithm that maintains a posterior over latent MDP parameters φ. However, the problem setting it considers and how it uses this posterior are quite different than what we consider in this paper.
>
> In this work, we are focused on zero-shot scenarios where the agent can only interact with the test MDP for a single episode; latent parameters are resampled for each episode. The PSRL regret analysis assumes MDPs with finite horizons and repeated episodes with the same test MDP, i.e. the latent parameters are fixed for all episodes.
>
> Before each episode, PSRL samples an MDP from its posterior over MDPs, computes the optimal policy for the sampled MDP, and executes it on the fixed test MDP. Its posterior is updated after each episode, concentrating the distribution around the true latent parameters. During this exploration period, it can perform arbitrarily poorly (see Section 6.1 of the appendix for a concrete example). Furthermore, sampling a latent MDP from the posterior determinizes the parameters; as a result, there is no uncertainty in the sampled MDP, and the resulting optimal policies that are executed will never take sensing actions. In this work, we have focused on Bayesian RL problems where sensing is critical to performance. BRPO, like other Bayesian RL algorithms, focuses on learning the Bayes-optimal policy during training, which can be used at test time to immediately explore and exploit in a new environment.
>
> We have run additional experiments to compare with PSRL. To make it handle the zero-shot scenario, we made PRSL sample from the posterior at every timestep and execute the corresponding optimal expert (aware of the penalties) . Since PSRL does not induce sensing, vanilla PSRL agent results in -124.4 ± 11.3, as it suffers from a large number of penalties for reaching incorrect goals. When we run PSRL with sensing probability 0.5, this results in 464.1 ± 5.5, with task completion rate of 94%. The task incompletion comes from belief occasionally not collapsing to one target. In comparison, our agent achieves an average return of 465.65 ± 4.35, completing 100% of the episodes, even with suboptimal experts unaware of the penalties.
>
> 2. Why use baseline with sensing probability 0.5? Other exploration methods should be compared.
> We have added Appendix 6.2 to include results with a different exploration method, and duplicated those results here for convenience.
>
> The ensemble we considered in Section 5 randomly senses with probability 0.5. A more effective sensing ensemble baseline policy could be designed manually, and used as the initial policy for the BRPO agent to improve on. Designing such a policy can be challenging: it requires either task-specific knowledge, or solving an approximate Bayesian RL problem. We bypass these requirements by using BRPO.
>
> On the Maze10 environment, we have found via offline tuning that a more effective ensemble baseline agent senses only for the first 150 of 750 timesteps. The average return is 416.3 ± 9.4, which outperforms the original ensemble baseline average return of 409.5 ± 10.8. However, this is still lower than the BRPO agent that starts with that original ensemble, which accumulated an average return of 465.7 ± 4.7. This trained BRPO agent also achieves a task completion rate of 100%, which is better than the 96.3% completed by the improved ensemble baseline. The performance gap comes from the suboptimality of the ensemble recommendation, as experts are unaware of the penalty for reaching incorrect goals.
>
> 3. No theoretical support.
> We show that the BRPO agent operates on its own MDP, which we refer to as Residual-MDP. Since this is an MDP, BRPO enjoys the theoretical guarantees provided by its underlying batch policy optimization algorithm. For example, if it runs TRPO, it inherits the same monotonic improvement guarantee.
>
> We have updated Section 4.2 to clarify this.
>
> [1] Osband, Ian, Daniel Russo, and Benjamin Van Roy. "(More) efficient reinforcement learning via posterior sampling." Advances in Neural Information Processing Systems. 2013.

---

### Official Review · AnonReviewer3 · 2019-10-29
**Official Blind Review #3**

**Rating:** 3

**Review:**

In this paper, the authors motivate and propose a learning algorithm, called Bayesian Residual Policy Optimization (BRPO), for Bayesian reinforcement learning problems. Experiment results are demonstrated in Section 5.

The paper is well written in general, and the proposed algorithm is also interesting. However, I think the paper suffers from the following limitations:

1) This paper does not have any theoretical analysis or justification. It would be much better if the authors can rigorously prove the advantages of BRPO under some simplifying assumptions.

2) It would be better if the authors can provide more experiment results, like experiment results in more games.

**Experience Assessment:**

I have published one or two papers in this area.

**Review Assessment: Checking Correctness Of Derivations And Theory:**

N/A

**Review Assessment: Checking Correctness Of Experiments:**

I assessed the sensibility of the experiments.

**Review Assessment: Thoroughness In Paper Reading:**

I read the paper at least twice and used my best judgement in assessing the paper.

---

> ### Author Response · Authors · 2019-11-15
> **Response**
>
> Thank you for your thoughtful comments and feedback. We have updated our paper to address your questions and comments. Here’s our summary.
>
> 1. No theoretical support
> We have updated our paper to highlight the theoretical contribution and to compare with other approaches that yield different theoretical guarantees. Please see the comments (1) and (3) under Reviewer 1 and the corresponding updated sections (Section 4.2, the Appendix).
>
> 2. More experiments
> We have added new experiments to compare against PSRL, as well as a more effective baseline ensemble policy as per Reviewer 1’s comment. We have run another experiment to handle continuous latent parameters, as pointed out by Reviewer 2. Please check the added Appendix, as well as our response (2) to Reviewer 1 and response (1) to Reviewer 2.

---

### Official Review · AnonReviewer2 · 2019-11-02
**Official Blind Review #2**

**Rating:** 6

**Review:**

The paper presents a Bayesian residual policy which improves a ensemble of expert policies by learning to reduce uncertainty. The algorithm is designed for reducing uncertainty due to the occluded objects and uncertainty about tasks. It is verified on two problems, cheese finding and door findiing, and compared with several different baselines.

The idea of the paper is good and Algorithm 1 sets out to learn the exploration policy when the expert policies do not agree. The exposition and writing are clear. The experiments are details and convey that the proposed method outperforms the baselines.

That said, the formulation of the task is a bit unusual and too specific, making me wonder if the method works for other tasks. Some questions to clarify the task formulation:
1. Do agent start locations and cheese locations change during the training and evaluation? The figures suggest they remain the same, in which case the generality is limited.

2. When an agent senses for cheese, does it receive orientation or only the distance? If it receives the distances, will that not be a signal that matches the goals with some noise. In other words, why does the agent to sense several times at the beginning to associate which expert policy should be active, and then follow that policy.

3. Why and how was the reward for the cheese finding task determined? It seems very specific.

4. I would be helpful to provide some intuition about \psi

Overall an interesting paper, but not sure how well it would perform on a wider set of tasks.

**Experience Assessment:**

I have read many papers in this area.

**Review Assessment: Checking Correctness Of Derivations And Theory:**

I assessed the sensibility of the derivations and theory.

**Review Assessment: Checking Correctness Of Experiments:**

I carefully checked the experiments.

**Review Assessment: Thoroughness In Paper Reading:**

I read the paper at least twice and used my best judgement in assessing the paper.

---

> ### Author Response · Authors · 2019-11-15
> **Reponse**
>
> Thank you for your thoughtful comments and feedback. We have updated our paper to address your questions and comments. Here’s our summary.
>
> 1. [Handling continuous latent parameters]
> The maze tasks in the submission consider only a finite number of latent goals. This is analogous to settings in which we have strong structural priors about where to search for the target, so the belief is represented as a categorical distribution over those candidates. However, our algorithm is not limited to such settings. We have run another experiment in which the goal is continuous and can be anywhere in the maze. The goal is tracked with an EKF, with mean and covariance. The expert recommendation is a motion planner path to the EKF’s mean goal. The BRPO agent results in average performance of 467.2, significantly outperforming BPO (152.7) or UPMLE (124.3). However, in this case, the BRPO agent does not improve significantly above the ensemble policy, which we believe is because it tracks only a unimodal belief. We plan to include experiments with multimodal belief representations, such as Gaussian Mixture Models, in our final submission.
>
> 2. The agent only receives noisy L2 distance to the goal. We intentionally constrained the observation because otherwise the goal becomes too obvious. This is a common setup in similar tasks (e.g. LightDark) explored in previous work [1].
>
> 3. The problem setup for Maze tasks is motivated by classical POMDP problems in discrete state-action spaces, such as Tiger [2] and RockSample [3]. Much like Maze tasks, these tasks also have a high reward for correct goal and a high penalty for incorrect goals, and low sensing costs. RockSample requires long-horizon navigation to get to the goals, which we also adopted.
>
>
> 4. φ is a latent parameter that drives the transition or reward functions. For a robot system, that could be uncertain parameters such as friction coefficients or joint damping parameters, or the mass of unknown objects being manipulated. For the Maze problems, it corresponds to the true location of the goal (cheese).
>
> [1] Robert Platt, Russ Tedrake, Leslie Pack Kaelbling, and Tomas Lozano-Perez. Belief space planning assuming maximum likelihood observations. In Robotics: Science and Systems, 2010.
> [2] Leslie Pack Kaelbling, Michael Littman, and Anthony Cassandra. Planning and acting in partially observable stochastic domains. Artificial Intelligence, 101(1-2):99–134, 1998.
> [3] Trey Smith and Reid Simmons. Heuristic search value iteration for pomdps. In Proceedings of the 20th conference on Uncertainty in artificial intelligence, pages 520–527. AUAI Press, 2004.

---

### Author Response · Authors · 2019-11-15
**Reponse**

We’d like to thank all reviewers for their thoughtful comments and feedback. We have updated our paper to address the questions and comments. Specific comments have been added to respond directly to each reviewer.

---

### Decision · Program_Chairs · 2019-12-19

**Decision:**

Reject

**Comment:**

This paper constitutes interesting progress on an important topic; the reviewers identify certain improvements and directions for future work (see in particular the updates from AnonReviewer1), and I urge the authors to continue to develop refinements and extensions.